# *TLR9* gene polymorphism -1237T/C (rs5743836) is associated with low IgG antibody response against *Pv*CSP variants in symptomatic *P. vivax* infections in Venezuela

Fhabián S. Carrión-Nessi[1,2,3], Eva Salazar-Alcalá[1], David A. Forero-Peña[2,3,4], Kellys A. Curiel[2], Mary Lopez-Perez[5], Mercedes Fernández-Mestre[1]*

**1** Immunogenetics Section, Laboratory of Pathophysiology, Centro de Medicina Experimental, Instituto Venezolano de Investigaciones Científicas, Altos de Pipe, Venezuela, **2** Biomedical Research and Therapeutic Vaccines Institute, Ciudad Bolívar, Venezuela, **3** "Luis Razetti" School of Medicine, Universidad Central de Venezuela, Caracas, Venezuela, **4** Department of Infectious Diseases, Hospital Universitario de Caracas, Caracas, Venezuela, **5** Center for Translational Medicine and Parasitology, Department of Immunology and Microbiology, Faculty of Health and Medical Sciences, University of Copenhagen, Copenhagen, Denmark

* mfernandezmestre@gmail.com

## Abstract

### Background

Clinical immunity to malaria has been associated with humoral immune responses targeting asexual-stage parasite proteins. However, the variability in antibody-driven responses may be influenced by genetic factors in the human host. This study aimed to evaluate the frequency of SNPs in the *TLR9* gene and their association with IgG antibody responses against *Pv*CSP variants (VK247, VK210, and V-like) and *Pv*MSP-$1_{19}$ among individuals presenting with symptomatic *Plasmodium vivax* infections in a malaria-endemic region of Venezuela.

### Methodology/Principal findings

A cross-sectional study was conducted in Bolívar state, Venezuela, involving 210 individuals infected with *P. vivax*. IgG reactivity against *P. vivax* recombinant antigens was assessed by ELISA, and three *TLR9* gene SNPs (rs5743836, rs352140, and rs187084) were genotyped by PCR. The median age of individuals was 29 years, with the majority being male miners with a prior history of malaria. Over 90% of individuals exhibited IgG antibodies against the three *Pv*CSP variants and *Pv*MSP-$1_{19}$. High responders to the *Pv*CSP variants reported fewer symptoms compared to medium ($p < 0.001$) and low ($p < 0.001$) responders. Among the analyzed SNPs, heterozygous genotypes were the most prevalent. Using an overdominant inheritance model, carrying the heterozygous genotype (T/C) for *TLR9* gene SNP rs5743836 was associated with lower IgG antibody response against *Pv*CSP VK247 (aOR = 0.26,

**Data availability statement:** All relevant data are within the manuscript and its Supporting information files.

**Funding:** The authors received no funding for this research.

**Competing interests:** I have read the journal's policy and the authors of this manuscript have the following competing interests: Mary Lopez-Perez is an Editorial Board Member of the journal PLOS Neglected Tropical Diseases. The rest of the authors have declared that no competing interests exist.

95% CI = 0.09 to 0.73, $p = 0.0094$) and *Pv*CSP V-like (aOR = 0.37, 95% CI = 0.14 to 0.99, $p = 0.047$). No significant associations between inheritance models for the three SNPs and parasitemia were observed.

## Conclusions/Significance

These findings suggest that *TLR9* gene variability, particularly the heterozygous genotype for SNP rs5743836, may influence the IgG antibody response against *Pv*CSP VK247 and V-like. Identifying genetic traits that impact immune response development could be valuable for malaria vaccine design and implementation.

### Author summary

In this study, we investigated how human genetic differences might affect the immune responses to malaria. Specifically, we looked at how certain genetic variations in the *TLR9* gene, which is part of the immune system, might affect the production of antibodies against malaria proteins in 210 Venezuelan individuals infected with *Plasmodium vivax*, one of the parasites that causes malaria. Most of these individuals were male miners likely infected in a highly endemic area. Our findings suggest that people with a specific genetic variation, known as a heterozygous genotype in the *TLR9* gene, had different levels of antibodies against two malaria proteins. Interestingly, those with higher antibody levels had fewer symptoms, suggesting that a stronger immune response may protect against malaria symptoms. Understanding these genetic influences on immune response could be valuable in designing vaccines and treatments that work more effectively in different populations.

## Introduction

*Plasmodium vivax* remains a leading public health problem in America, particularly in rural areas of Venezuela, Brazil, and Colombia [1]. Since 2016, Venezuela has reported the highest number of malaria cases and deaths in the region [1], with the southeastern part of the country, bordering Brazil and Guyana, remaining at high risk for malaria transmission [2]. There, Bolívar state accounts for 60–88% of the country's malaria cases and has become the hottest malaria hotspot in America [3].

The host immune response to *P. vivax* is triggered by various parasitic components, resulting in a synergistic interaction between innate and adaptive immunity to control and eliminate the parasite [4]. This response begins with the bite of an infected female *Anopheles* mosquito, which inoculates sporozoites into the host's skin, and develops through both hepatic and blood schizogony processes. The parasite's blood stages contain pathogen-associated molecular patterns, such as glycosylphosphatidylinositol anchors, hemozoin, and parasite nucleic acids, which trigger an innate immune response via diverse pattern recognition receptors, including

Toll-like receptors (TLRs) [5]. To date, 10 TLRs have been identified in humans, located either on the cell surface (TLR1, TLR2, TLR4, TLR5, TLR6, and TLR10) or on endosomes (TLR3, TLR7, TLR8, and TLR9) [6].

TLR9 participates in microbial DNA recognition, specifically unmethylated CpG motifs, and is expressed on professional antigen-presenting cells, such as dendritic cells and B cells [6]. Additionally, TLR9 regulates B cell viability, apoptosis, and antibody and IL-10 production by upregulating the Fc receptor-like 3 [7]. The activation of memory B cells by TLR9 leads to cell expansion and immunoglobulin production, which may be critical for memory B cell homeostasis and sustained antibody production [8]. In malaria, TLR9 recognizes free or hemozoin-bound parasite DNA [5,9]. Indeed, the high content of CpG motifs in *P. vivax* is likely responsible for the species' high fever-inducing capacity [10].

In malaria-endemic areas, particularly where transmission rates are high, individuals become less susceptible to symptomatic malaria episodes with increasing age and exposure [11]. The role of antibodies in malaria protection is well documented [12], and the host's genetic makeup is known to significantly influence the development of an effective immune response against the parasite [13]. Compared to several studies in *P. falciparum* [14], only two studies have reported *TLR9* gene polymorphisms in *P. vivax* [15,16]. Moreover, only a few studies have documented the role of immune-related gene polymorphisms on antibody production against *P. vivax* proteins [17–21]. To date, no study has evaluated the effect of single nucleotide polymorphisms (SNPs) of the *TLR9* gene on the IgG antibody response against circumsporozoite protein (*Pv*CSP), and only one study has done so against merozoite surface protein 1 (*Pv*MSP-1$_{19}$) [16].

Here, we evaluated the frequency of *TLR9* gene SNPs rs5743836 (-1237T/C), rs352140 (1635A/G), and rs187084 (-1486C/T) and their associations with IgG antibody response against *Pv*CSP variants (VK247, VK210, and V-like) and *Pv*MSP-1$_{19}$ among individuals with symptomatic *P. vivax* infections in Bolívar state, Venezuela.

## Methods

### Ethical approval and consent to participate

This study adhered to the ethical guidelines established by national and institutional committees overseeing human experimentation following the ethical principles of the Declaration of Helsinki. The study protocol was reviewed and approved by the Independent Bioethics Committee for Research of the Bioethical National Center of Venezuela (Approval No.: CIBI-CENABI-03/2022). All patients received a comprehensive explanation of the study design and the data and sample collection methodology. Only those who voluntarily agreed to participate and provided signed consent were included in the study.

### Study area

This study was conducted in the Laboratory of Malariology of the "Ruiz y Páez" University Hospital Complex (Bolívar state), which serves as the main reference center for diagnosis and treatment of malaria patients in the state. Bolívar state, situated in the southeastern part of the country, is Venezuela's largest political and administrative region, with an estimated population of over 1,947,403 [22]. Most of the population resides in the northern and eastern areas of the state, where primary economic activities include mining, steel and aluminum manufacturing, hydroelectric power generation, commerce and services, forestry, livestock rearing, and agricultural development [3].

According to the most recent official publication on the national epidemiological situation of malaria in Venezuela (Week No. 41, 2022) [23], a total of 60,329 accumulated cases had been reported in Bolívar state, representing 59% of malaria cases in Venezuela. *P. vivax* (78%) was the dominant parasite species, followed by *P. falciparum* (16%) and mixed (*P. vivax*/*P. falciparum*) infection (6%) [23].

### Study design and population

A cross-sectional study was conducted between June 2022 and January 2023 involving unrelated Venezuelan individuals infected with *P. vivax*, either primo-infected or with previous infections, who presented with malaria-compatible symptoms at the Laboratory of Malariology of the "Ruiz y Páez" University Hospital Complex. All individuals included in the study

were classified as uncomplicated malaria cases according to the Venezuelan criteria [24]. The diagnosis of *Plasmodium* spp. was performed by thick and thin blood smears stained with the Giemsa method according to the WHO guidelines [25]. For quality assurance, each slide was independently examined by two experienced microscopists using 100X oil immersion magnification. Both microscopists identified *Plasmodium* spp. and quantified parasitemia from the thick blood smear [26]. This diagnosis was subsequently confirmed by nested PCR [27] at the Laboratory of Pathophysiology (Immunogenetics Section) of the Instituto Venezolano de Investigaciones Científicas. Individuals with mixed (*P. vivax*/*P. falciparum* or other species) infection, those undergoing antimalarial treatment at the time of blood sample collection, immunocompromised individuals (e.g., transplant recipients or chemotherapy patients), and those with other self-reported infectious, autoimmune, or hematologic diseases (e.g., sickle cell disease), were not enrolled.

Additionally, 40 healthy individuals without previous malaria exposure (negative control samples) from non-endemic areas and no history of blood transfusion were included as controls. This control group was used in the ELISA assays to establish the cut-off point necessary to determine the antibody reactivity index (RI).

### Blood sample collection and genomic DNA extraction

After applying a clinical-epidemiological data collection form, two blood samples were collected from each individual by peripheral venipuncture. One sample was collected in a tube without anticoagulant to obtain serum and a second with EDTA for molecular analysis. The genomic DNA was extracted from whole blood using the protocol described by Bunce [28]. The concentration and purity of the genomic DNA were quantified by ultraviolet spectrophotometry using the Nano-Drop 2000 (Thermo Fisher Scientific, Waltham, MA, USA). The quality of the genomic DNA was verified by agarose gel electrophoresis.

### Recombinant antigens

Recombinant $PvMSP-1_{19}$ (amino acids 1616–1704, Belém strain) was expressed in *Escherichia coli* with a polyhistidine affinity tag (6xHis tag), as previously described [29], at the Department of Molecular Biology and Immunology of the Fundación Instituto de Inmunología de Colombia. Peptides corresponding to the three variants (VK247, VK210, and V-like) of the *Pv*CSP repetitive region were synthesized in Fmoc solid-phase [30] at the Instituto de Medicina Tropical (Laboratory of Peptide Synthesis) of the Universidad Central de Venezuela. Sequences are shown in S1 Table.

### Evaluation of IgG antibody response against the three *Pv*CSP variants and $PvMSP-1_{19}$

IgG reactivity against *P. vivax* recombinant antigens was determined by ELISA. All steps were performed at room temperature in a humidity chamber. Briefly, 96-well flat-bottom plates (Nunc MaxiSorp ELISA, Thermo Fisher Scientific, Waltham, MA, USA) were coated for 2 hours with *Pv*CSP variants (5 µg/mL) or $PvMSP-1_{19}$ (10 µg/mL) diluted in carbonate/bicarbonate pH 9.6. After washing three times with 0.05% Tween 20 in phosphate-buffered saline (PBS-T), the plates were blocked for 30 minutes with 1% bovine serum albumin (BSA) in PBS. Then, 100 µL of diluted serum (1:200 for *Pv*CSP variants and 1:100 for $PvMSP-1_{19}$ in 0.5% BSA in PBS-T) were added to triplicate wells and incubated for 1 hour. After washing, 1:10,000 horseradish peroxidase-conjugated anti-human IgG (Santa Cruz Biotechnology, Dallas, TX, USA) was added to each well and incubated for 1 hour. Bound antibodies were detected by adding TMB (Sigma-Aldrich, Burlington, MA, USA) and the reaction was stopped with 0.1 M sulfuric acid. The absorbance was read on an Infinite M200 ELISA plate reader (Tecan, Männedorf, Switzerland) at 405 nm.

IgG antibody response against *P. vivax* recombinant antigens was expressed as a RI, which was determined by dividing the optical density (OD) value of the test sample by the cut-off point OD value. The cut-off values were calculated by averaging the OD values plus three standard deviations obtained with negative control samples. Samples from individuals with an RI ≥ 1 (also known as responders) were considered positive.

## Genotyping of *TLR9* gene SNPs

The SNP rs5743836 was determined by bi-directional PCR amplification of specific alleles, using previously described primers [31]. For this, 1 µL of DNA was mixed with 1X Tonbo VerityMAX DNA polymerase master mix, 0.4 µM primer P, 0.4 µM primer Q, 0.05 µM primer W, and 0.1 µM primer M. Amplified products were separated by electrophoresis in a 1.5% agarose gel and visualized using a ChemiDoc MP imaging system (Bio-Rad Laboratories, Hercules, CA, USA). Depending on the genotype, two or three overlapping fragments were produced. PQ primers served as the internal control (644 bp), PW primers amplified the T allele (395 bp), and MQ primers amplified the C allele (275 bp) (S2 Table).

SNPs rs352140 and rs187084 were determined by PCR-restriction fragment length polymorphism, using previously described primers and protocols [32]. Here, 1 µL of DNA was mixed with 1X Tonbo VerityMAX DNA polymerase master mix, 1 pM forward and 1 pM reverse primers. Amplified products were digested with restriction enzymes (New England Biolabs, Ipswich, MA, USA) according to the manufacturer's instructions, separated by electrophoresis in a 2% agarose gel, and visualized using a ChemiDoc MP imaging system. Genotypes were assigned based on the number and size of fragments resulting from the digestion of the amplified product (S2 Table).

## Statistical analysis

Data were analyzed using SPSS version 27 (IBM Corp., Armonk, NY, USA) and plotted using GraphPad Prism version 10.1.2 (GraphPad Software, Boston, MA, USA). Individual data were summarized using descriptive statistics. A two-stage cluster analysis was performed to identify natural groupings (or clusters) of parasitemia and RI against the three *Pv*CSP variants and *Pv*MSP-1$_{19}$. The method used a Log-likelihood distance measure and involved an initial pre-clustering step creating a Cluster Feature tree, followed by an agglomerative hierarchical clustering stage on the subclusters. The optimal number of clusters was determined automatically by comparing potential solutions using the Schwarz Bayesian Information Criterion or the Akaike Information Criterion. The resulting clusters were examined based on their mean RI profiles and subsequently designated as "low", "medium", and "high" responder groups. The distribution of numerical variables was assessed using the Kolmogorov-Smirnov test, and one-way ANOVA or Kruskal-Wallis tests were applied as appropriate. Pearson's chi-square and Fisher's exact tests were used for categorical variables. Genetic analyses were conducted using the SNPStats program [33]. Allele, genotype, and haplotype frequencies, Hardy-Weinberg equilibrium, linkage disequilibrium, and inheritance models were analyzed. Multiple regression models adjusted for age, sex, mining occupation, probable area of infection, and history of malaria were performed to evaluate the effect of potential confounders on the association between IgG antibody response level and parasitemia, and *TLR9* gene SNPs. A $p$ value $< 0.05$ was considered statistically significant.

## Results

### Clinical-epidemiological characteristics of the study population

A total of 210 unrelated Venezuelan individuals with symptomatic malaria by *P. vivax* were included in the analysis (Fig 1). The median age of the individuals was 29 (IQR 21–43) years, most were male (60.5%, $n = 127$), and practiced illegal gold mining (56.2%, $n = 118$). The probable area of infection for all individuals was Bolívar state, primarily in Sifontes (38.1%, $n = 80$), Sucre (26.7%, $n = 56$), and Angostura del Orinoco (22.4%, $n = 47$) municipalities. More than 60% of individuals had low parasitemia (≤ 4800 parasites/µL of blood), and over 80% reported a history of malaria, with a median of 5 (IQR 2–12) episodes (Table 1).

All participants reported at least one symptom, with a median number of 7 (IQR 5–10). The most common symptoms were fever (94.3%, $n = 198$), headache (92.4%, $n = 194$), chills (76.7%, $n = 161$), low back pain (71.9%, $n = 151$), and arthralgia (68.6%, $n = 144$).

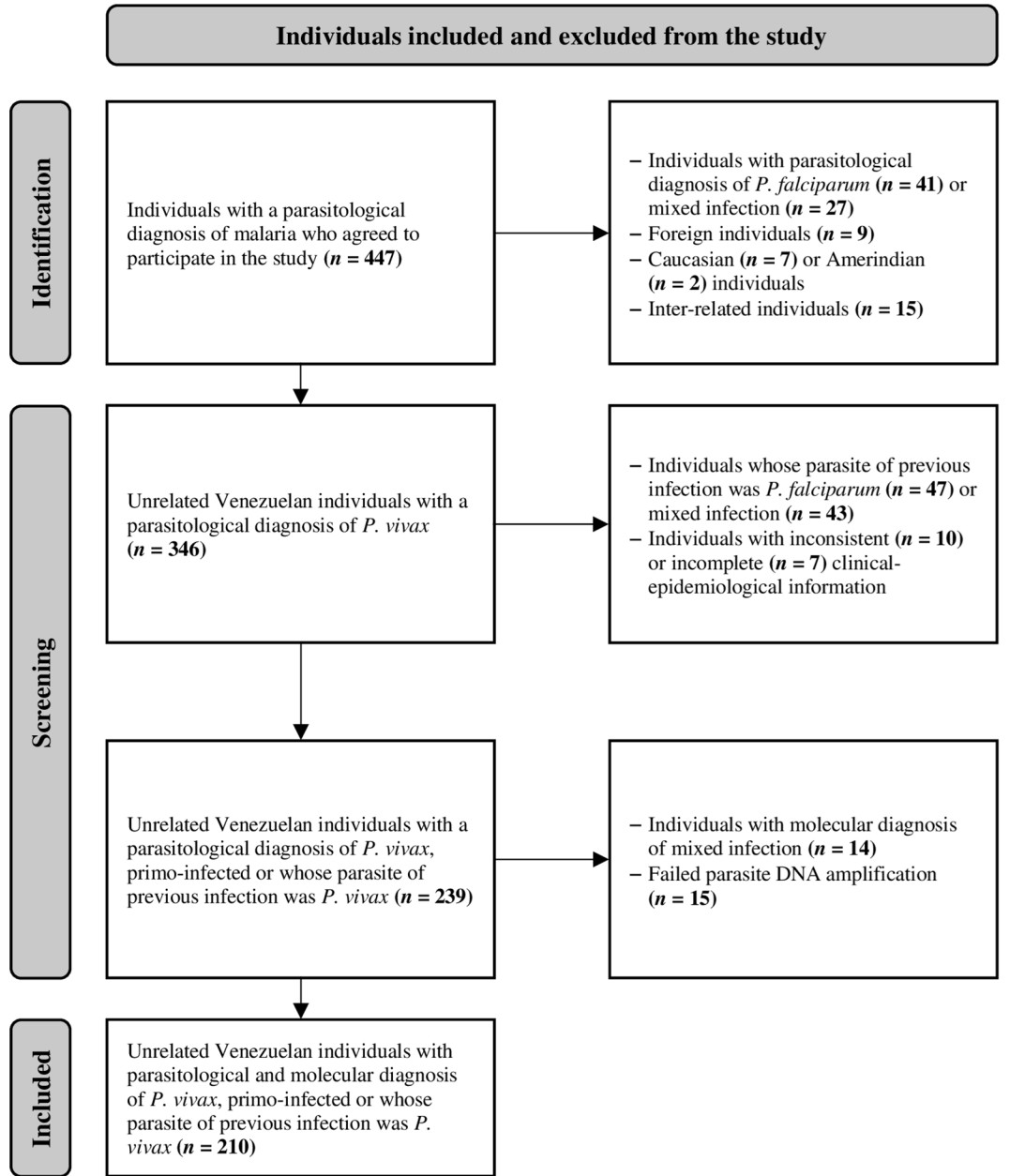

**Fig 1. Flow diagram illustrating identification, screening, and inclusion of 210 participants.** Individuals who self-identified mainly as Caucasian (*n*=7, primary European ancestry) or Amerindian (*n*=2, Venezuelan Indigenous ethnic group) were excluded to minimize confounding factors related to population stratification in the genetic association study of *TLR9* gene SNPs. This focused the analysis on the general admixed population ("criollo"/ "mestizo") characteristic of the study area in Bolívar state.

### High seroprevalence of *Pv*CSP and *Pv*MSP-1$_{19}$-specific IgG

More than 90% of individuals had IgG antibodies (RI ≥ 1) against the three *Pv*CSP variants and *Pv*MSP-1$_{19}$ (Fig 2). Using a two-stage cluster analysis, we grouped individuals into low, medium, and high responders according to their IgG antibody response levels. This classification was used for following analyses. No statistically significant differences were

PLOS Neglected Tropical Diseases

**Table 1. Clinical-epidemiological characteristics of individuals infected with *P. vivax*.**

| Clinical-epidemiological characteristics | *n* = 210 |
|---|---|
| Age, median (IQR), years | 29 (21-43) |
| Sex, male (%) | 127 (60.5) |
| Occupation, *n* (%) | |
| Illegal gold mining | 118 (56.2) |
| Homemaker | 28 (13.3) |
| Farmer | 15 (7.1) |
| Others | 49 (23.4) |
| PAI (municipality), *n* (%) | |
| Sifontes | 80 (38.1) |
| Sucre | 56 (26.7) |
| Angostura del Orinoco | 47 (22.4) |
| Others | 27 (12.8) |
| Parasitemia, median (IQR),/µL | 4,300 (3,500-5,800) |
| Previous malaria, yes (%) | 170 (81) |
| No. of total episodes, median (IQR) | 5 (2-12) |
| No. of episodes in the last year, median (IQR) | 2 (1-5) |
| Days since last episode, median (IQR) | 87.5 (57-129) |

IQR: interquartile range; PAI: probable area of infection.

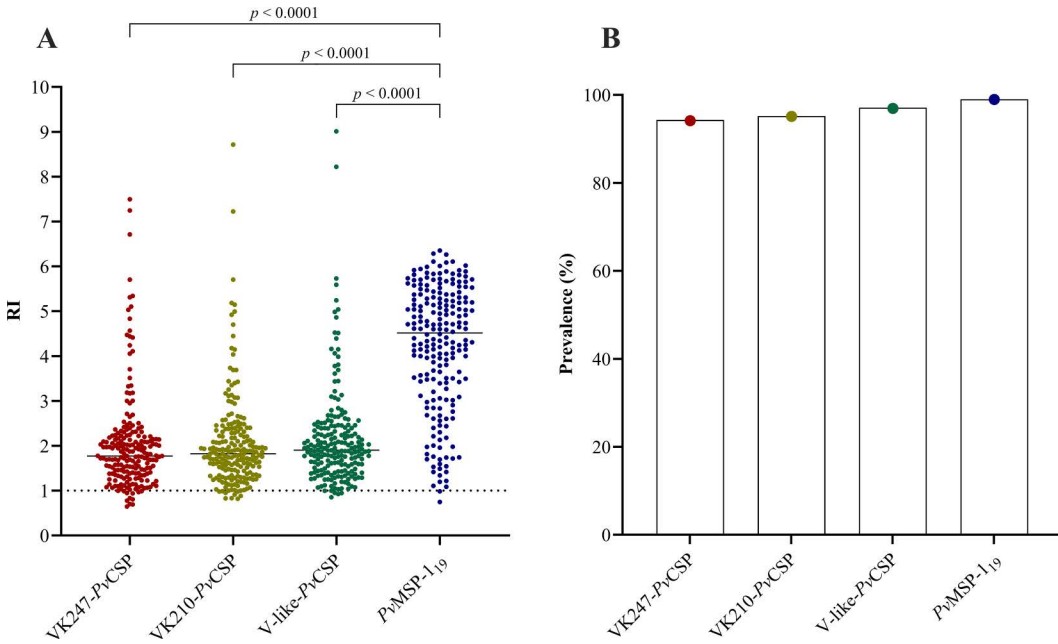

**Fig 2. IgG levels against the three *Pv*CSP variants and *Pv*MSP-1₁₉. (A)** Specific IgG levels against recombinant *Pv*CSP and *Pv*MSP-1$_{19}$ in serum from adults (*n* = 210) determined by ELISA. Values are expressed as reactivity index (RI). Medians and *p* values are shown using the Kruskal-Wallis test followed by Dunn's multiple comparison test. Negative cut-off values (RI = 1) are shown as dotted lines and were used to calculate seroprevalence. **(B)** Seroprevalence of *Pv*CSP- and *Pv*MSP-1$_{19}$-specific IgG.

observed between the clinical-epidemiological characteristics of the individuals and their IgG antibody response levels against the three *Pv*CSP variants (S3–S5 Tables). However, a higher proportion of previous malaria was observed in high responders against *Pv*MSP-1$_{19}$ compared to low responders (85.7% vs. 68.1%, $p = 0.013$) (S6 Table).

High responders against the three *Pv*CSP variants had a lower number of symptoms compared to medium ($p < 0.001$) and low ($p < 0.001$) responders (Fig 3).

### *TLR9* gene SNP rs5743836 (-1237T/C) is associated with low IgG antibody response against *Pv*CSP VK247 and V-like

Genotypic and allelic frequencies of *TLR9* gene SNPs are summarized in Fig 4. Heterozygous genotypes T/C (66.7%, $n = 140$), A/G (57.1%, $n = 120$), and C/T (81.4%, $n = 171$) for SNPs rs5743836, rs352140, and rs187084, respectively, were the most frequent among 210 analyzed individuals. For SNP rs187084, the homozygous genotype for the minor allele (T/T) was not observed. None of the analyzed *TLR9* gene SNPs conformed to Hardy-Weinberg equilibrium ($p < 0.05$). Moderate linkage disequilibrium (D'= 0.512, $r = 0.47$) was observed between SNPs rs5743836 and rs187084 ($p < 0.001$), and strong linkage disequilibrium (D'= 0.753, $r = 0.64$) was observed between SNPs rs352140 and rs187084 ($p < 0.001$).

Then, we evaluated the effects of *TLR9* gene SNPs on the IgG antibody response, comparing low and high responders. According to the best-fit inheritance model (overdominant), carrying the heterozygous genotype (T/C) for the SNP rs5743836 of the *TLR9* gene was associated with low IgG antibody response against *Pv*CSP VK247 (aOR = 0.26, 95% CI = 0.09 to 0.73, $p = 0.0094$) (Table 2) and *Pv*CSP V-like (aOR = 0.37, 95% CI = 0.14 to 0.99, $p = 0.047$) (Table 3). No other significant differences were observed between SNP rs5743836 inheritance models and IgG antibody response level against the *Pv*CSP VK210 (S7 Table) or *Pv*MSP-1$_{19}$ (S8 Table). Likewise, no differences were observed between SNP rs352140 inheritance models and IgG antibody response level against the three *Pv*CSP variants (Tables 2, 3 and S7) and or against *Pv*MSP-1$_{19}$ (S8 Table).

Haplotype analysis revealed that the combination of major alleles of the SNPs rs5743836, rs352140, and rs187084 of the *TLR9* gene (TAC) was the most common among the individuals. Other haplotypes were observed with variable distribution (S9 Table). CAT and CGC haplotypes were not observed in high responders against *Pv*CSP VK247, and *Pv*CSP VK210 and V-like, respectively. Despite some trends, no significant differences in haplotypic frequencies were observed between low and high responders (S9 Table).

No significant differences between inheritance models for the SNPs rs5743836 and rs352140 and parasitemia were observed (Table 4). Due to the absence of the homozygous genotype for the minor allele (T/T), it was not possible to establish inheritance models for the SNP rs187084.

## Discussion

Innate and adaptive mechanisms of the immune system have evolved to adapt and control the pathogenesis of infectious diseases and to repair tissues [34]. The pathogen-host interaction in *P. vivax* malaria is biologically complex and influenced by environmental factors, parasitemia, immune status, and the genetic makeup of the human host [35]. Whereas most of the studies have focused on *P. falciparum*, we explored here the effects of *TLR9* gene polymorphisms on the IgG antibody response against two malaria antigens, *Pv*CSP and *Pv*MSP-1$_{19}$.

As previously reported in Bolívar state [36,37], most individuals were young adult males engaged in illegal gold mining. Over 80% of individuals reported previous malaria episodes, with a median of five episodes, highlighting the importance of the state as a hotspot in Latin America [3]. In this study, the parasitemia distribution reflects the reported in Colombia [38], but appears constrained compared to broader ranges reported in the Asia-Pacific region [39,40]. This difference is probably due to the symptomatic status of all participants and a single-timepoint sampling, potentially missing wider density fluctuations [41,42]. In this study, over 90% of individuals had IgG antibodies (RI ≥ 1) against *Pv*CSP or *Pv*MSP-1$_{19}$, similar to other studies conducted in Latin America [21,43,44]. Although no significant differences were observed between

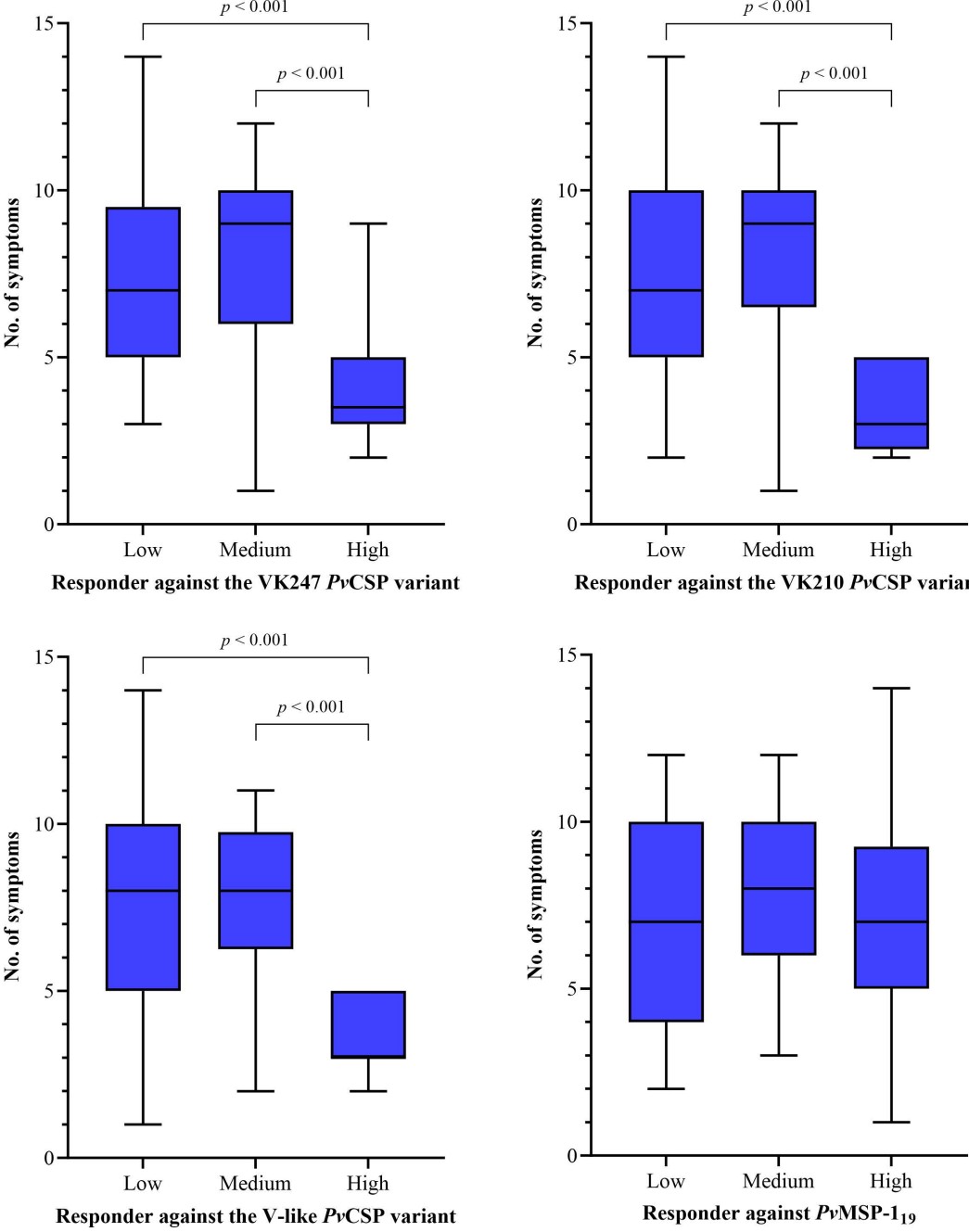

**Fig 3. High responders against *Pv*CSP have a lower number of symptoms.** Number of symptoms among low, medium, and high responders against the VK247 (*n* = 198), VK210 (*n* = 200), V-like (*n* = 204), and *Pv*MSP-1$_{19}$ (*n* = 208). Responders were defined by a two-stage cluster analysis using their reactivity index (RI) for each tested antigen. Median, IQR, and whiskers (1.5 times the IQR) are shown. *P* values using the Kruskal-Wallis test followed by Dunn's multiple comparison test are also shown.

the clinical-epidemiological characteristics of individuals according to their specific-IgG response, male miners were commonly high responders. This agrees with previous studies showing an association between mining activity and high malaria transmission in this area [3,45].

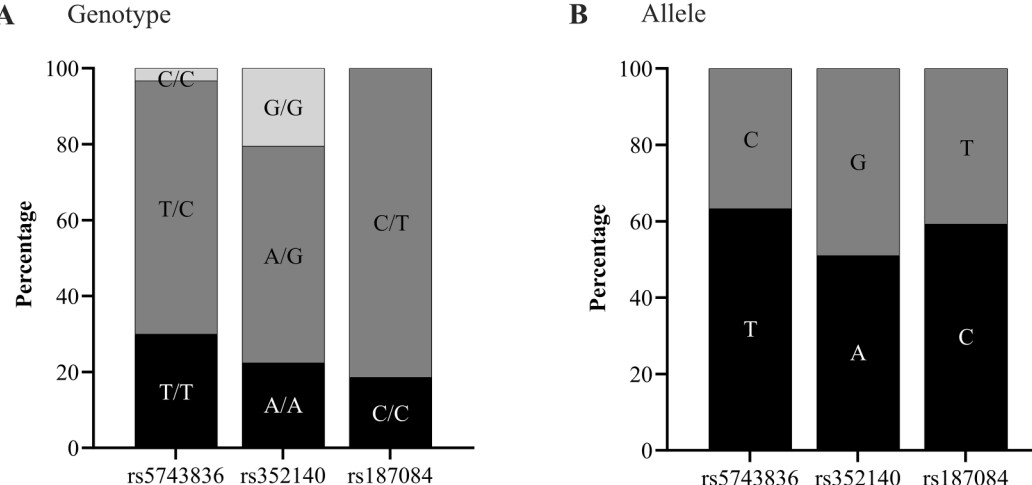

**Fig 4. Genotypic and allelic frequencies for SNPs rs5743836, rs352140, and rs187084 of the *TLR9* gene.** Proportion of individuals infected with *P. vivax* (*n* = 210) carrying a specific genotype (**A**) or allele (**B**) for SNPs rs5743836, rs352140, and rs187084 of the *TLR9* gene.

**Table 2. Association of genotypic frequencies of *TLR9* gene SNPs with IgG antibody response level against *Pv*CSP VK247 .**

| Inheritance models | Genotypes | Responder against *Pv*CSP VK247 | | aORᵃ (95% CI) | p value | AIC |
|---|---|---|---|---|---|---|
| | | Low (*n* = 179, 90.4%) | High (*n* = 19, 9.6%) | | | |
| rs5743836 | | | | | | |
| Codominant | T/T | 46 (25.7) | 11 (57.9) | 1 | 0.034 | 137.8 |
| | T/C | 128 (71.5) | 7 (36.8) | 0.26 (0.09-0.76) | | |
| | C/C | 5 (2.8) | 1 (5.3) | 1.26 (0.1-15.85) | | |
| Dominant | T/T | 46 (25.7) | 11 (57.9) | 1 | 0.018 | 137 |
| | T/C-C/C | 133 (74.3) | 8 (42.1) | 0.29 (0.11-0.81) | | |
| Recessive | T/T-T/C | 174 (97.2) | 18 (94.7) | 1 | 0.48 | 142 |
| | C/C | 5 (2.8) | 1 (5.3) | 2.64 (0.22-32.07) | | |
| Overdominant | T/T-C/C | 51 (28.5) | 12 (63.2) | 1 | 0.009 | 135.8 |
| | T/C | 128 (71.5) | 7 (36.8) | 0.26 (0.09-0.73) | | |
| Additive | – | – | – | 0.38 (0.14-1) | 0.048 | 138.7 |
| rs352140 | | | | | | |
| Codominant | A/A | 41 (22.9) | 3 (15.8) | 1 | 0.83 | 144.2 |
| | A/G | 103 (57.5) | 13 (68.4) | 1.47 (0.38-5.77) | | |
| | G/G | 35 (19.6) | 3 (15.8) | 1.2 (0.21-6.88) | | |
| Dominant | A/A | 41 (22.9) | 3 (15.8) | 1 | 0.6 | 142.3 |
| | A/G-G/G | 138 (77.1) | 16 (84.2) | 1.41 (0.37-5.4) | | |
| Recessive | A/A-A/G | 144 (80.5) | 16 (84.2) | 1 | 0.85 | 142.5 |
| | G/G | 35 (19.6) | 3 (15.8) | 0.88 (0.23-3.43) | | |
| Overdominant | A/A-G/G | 76 (42.5) | 6 (31.6) | 1 | 0.57 | 142.2 |
| | A/G | 103 (57.5) | 13 (68.4) | 1.35 (0.47-3.9) | | |
| Additive | – | – | – | 1.1 (0.48-2.49) | 0.82 | 142.5 |

ᵃMultiple logistic regression model comparing low to high responders was used to calculate odds ratios (ORs), 95% confidence intervals (CI), and *p* values. The model was adjusted for age, sex, mining occupation, probable area of infection, previous malaria, number of total episodes, and days since the last episode. aOR: adjusted odds ratio; CI: confidence interval; AIC: Akaike information criterion.

Table 3. Association of genotypic frequencies of *TLR9* gene SNPs with IgG antibody response level against *Pv*CSP V-like .

| Inheritance models | Genotypes | Responder against *Pv*CSP V-like | | aORᵃ (95% CI) | p value | AIC |
|---|---|---|---|---|---|---|
| | | Low (n = 183, 89.7%) | High (n = 21, 10.3%) | | | |
| rs5743836 | | | | | | |
| Codominant | T/T | 50 (27.3) | 10 (47.6) | 1 | 0.11 | 150.2 |
| | T/C | 128 (70) | 9 (42.9) | 0.41 (0.15-1.14) | | |
| | C/C | 5 (2.7) | 2 (9.5) | 2.03 (0.28-14.6) | | |
| Dominant | T/T | 50 (27.3) | 10 (47.6) | 1 | 0.14 | 150.5 |
| | T/C-C/C | 133 (72.7) | 11 (52.4) | 0.47 (0.18-1.27) | | |
| Recessive | T/T-T/C | 178 (97.3) | 19 (90.5) | 1 | 0.22 | 151.1 |
| | C/C | 5 (2.7) | 2 (9.5) | 3.57 (0.54-23.6) | | |
| Overdominant | T/T-C/C | 55 (30.1) | 12 (57.1) | 1 | 0.047 | 148.7 |
| | T/C | 128 (70) | 9 (42.9) | 0.37 (0.14-0.99) | | |
| Additive | – | – | – | 0.69 (0.28-1.71) | 0.42 | 152 |
| rs352140 | | | | | | |
| Codominant | A/A | 42 (22.9) | 5 (23.8) | 1 | 0.75 | 154.1 |
| | A/G | 104 (56.8) | 13 (61.9) | 0.91 (0.29-2.86) | | |
| | G/G | 37 (20.2) | 3 (14.3) | 0.56 (0.11-2.84) | | |
| Dominant | A/A | 42 (22.9) | 5 (23.8) | 1 | 0.74 | 152.6 |
| | A/G-G/G | 141 (77) | 16 (76.2) | 0.82 (0.27-2.52) | | |
| Recessive | A/A-A/G | 146 (79.8) | 18 (85.7) | 1 | 0.46 | 152.1 |
| | G/G | 37 (20.2) | 3 (14.3) | 0.6 (0.15-2.43) | | |
| Overdominant | A/A-G/G | 79 (43.2) | 8 (38.1) | 1 | 0.78 | 152.6 |
| | A/G | 104 (56.8) | 13 (61.9) | 1.15 (0.43-3.09) | | |
| Additive | – | – | – | 0.77 (0.36-1.66) | 0.51 | 152.2 |

ᵃMultiple logistic regression model comparing low to high responders was used to calculate odds ratios (ORs), 95% confidence intervals (CI), and *p* values. The model was adjusted for age, sex, mining occupation, probable area of infection, previous malaria, number of total episodes, and days since the last episode. aOR: adjusted odds ratio; CI: confidence interval; AIC: Akaike information criterion.

High responders against *Pv*CSP variants exhibited fewer malaria symptoms than medium and low responders. Specific-*Pv*CSP antibodies block the invasion and development of sporozoites in human hepatocytes [46], which may prevent the development of blood stages and, in turn, clinical malaria. A higher proportion of previous malaria was observed in high responders against *Pv*MSP-1$_{19}$ compared to low responders, confirming a potentiating effect on the antibody response against this protein in individuals previously exposed to malaria [21], and suggesting an increased number of antibody-producing short-lived plasma cells, likely due to the presence of a memory B cell response, which rapidly proliferates and differentiates upon reinfection [47].

TLRs play a critical role in immune protection and have been subjected to selection in response to immune challenges, as observed in studies associating TLRs and their SNPs with disease [6]. To ensure the representativeness of our findings, SNPs were selected based on available data from genomic databases, particularly those focused on Latin American populations. Additionally, the chosen SNPs were selected based on prior research demonstrating significant associations with the diseases under investigation, with known functional effects. In this study, heterozygous genotypes of SNPs rs5743836, rs352140, and rs187084 of the *TLR9* gene were most frequent among evaluated individuals, as reported in African individuals [48] and among three United States ethnic groups (African American, European, and Hispanic) [49]. High heterozygosity has also been reported at SNPs rs352140 and rs187084 [15,50,51] or only rs5743836 [52]. These genetic differences across populations, as a result of balanced selection [53], complicate comparisons between studies conducted in different locations [14].

PLOS Neglected Tropical Diseases

**Table 4. Association of genotypic frequencies of *TLR9* gene SNPs with parasitemia.**

| Inheritance models | Genotypes | Low parasitemia (n=129, 61.4%) | High parasitemia (n=81, 38.6%) | aOR[a] (95% CI) | p value | AIC |
|---|---|---|---|---|---|---|
| rs5743836 | | | | | | |
| Codominant | T/T | 36 (27.9) | 27 (33.3) | 1 | 0.51 | 286.8 |
| | T/C | 87 (67.4) | 53 (65.4) | 0.94 (0.49-1.8) | | |
| | C/C | 6 (4.7) | 1 (1.2) | 0.29 (0.03-2.9) | | |
| Dominant | T/T | 36 (27.9) | 27 (33.3) | 1 | 0.74 | 286 |
| | T/C-C/C | 93 (72.1) | 54 (66.7) | 0.9 (0.47-1.72) | | |
| Recessive | T/T-T/C | 123 (95.3) | 80 (98.8) | 1 | 0.25 | 284.8 |
| | C/C | 6 (4.7) | 1 (1.2) | 0.31 (0.03-2.91) | | |
| Overdominant | T/T-C/C | 42 (32.6) | 28 (34.6) | 1 | 0.94 | 286.1 |
| | T/C | 87 (67.4) | 53 (65.4) | 1.02 (0.54-1.94) | | |
| Additive | – | – | – | 0.82 (0.46-1.48) | 0.51 | 285.7 |
| rs352140 | | | | | | |
| Codominant | A/A | 24 (18.6) | 23 (28.4) | 1 | 0.17 | 284.6 |
| | A/G | 77 (59.7) | 43 (53.1) | 0.52 (0.25-1.09) | | |
| | G/G | 28 (21.7) | 15 (18.5) | 0.47 (0.19-1.2) | | |
| Dominant | A/A | 24 (18.6) | 23 (28.4) | 1 | 0.06 | 282.6 |
| | A/G-G/G | 105 (81.4) | 58 (71.6) | 0.51 (0.25-1.03) | | |
| Recessive | A/A-A/G | 101 (78.3) | 66 (81.5) | 1 | 0.45 | 285.6 |
| | G/G | 28 (21.7) | 15 (18.5) | 0.75 (0.34-1.62) | | |
| Overdominant | A/A-G/G | 52 (40.3) | 38 (46.9) | 1 | 0.31 | 285.1 |
| | A/G | 77 (59.7) | 43 (53.1) | 0.73 (0.4-1.34) | | |
| Additive | – | – | – | 0.67 (0.42-1.08) | 0.09 | 283.4 |

[a]Multiple logistic regression model comparing low to high responders was used to calculate odds ratios (ORs), 95% confidence intervals (CI), and p values. The model was adjusted for age, sex, mining occupation, probable area of infection, previous malaria, number of total episodes, and days since the last episode. aOR: adjusted odds ratio; CI: confidence interval; AIC: Akaike information criterion

After controlling for potential biases that could influence the IgG antibody response, we found that the heterozygous genotype (T/C) for SNP rs5743836 of the *TLR9* gene was associated with low IgG antibody response against *Pv*CSP VK247 and V-like in the evaluated individuals. Findings on the promoter activity for SNP rs5743836 of the *TLR9* gene are contradictory. Studies have reported higher promoter activity in the homozygous genotype for the major allele (T/T) [54], but also for the minor allele (C/C) [55,56], and the heterozygous genotype (T/C) [57,58]. These discrepancies could be explained by differences in the etiologies of the models studied, the timing of transcriptional evaluation, and/or the presence of other functional SNPs that interact with rs5743836, regulating TLR9 expression [55,56]. Therefore, it may be deduced that in individuals carrying the heterozygous genotype for SNP rs5743836, *TLR9* gene transcription and B cell activation are dysregulated, affecting the IgG antibody response against *Pv*CSP. Notably, the overall clinical presentation and the specific IgG response in *P. vivax* malaria are shaped by a complex interplay of such host factors. Beyond TLRs, these encompass gene polymorphisms in cytokine [59–61], B-cell co-stimulatory molecules [21], human leukocyte antigens molecules [19,62], and Fc gamma receptors [63], alongside factors like active malaria infection status [64], prior exposure, and inherent *Pv*CSP immunogenicity [65–67]. However, current scientific evidence on the precise contributions and interactions of these factors is limited and often inconsistent, highlighting the need for comprehensive studies across diverse geographic regions and genetic background [14].

We did not observe an association between parasitemia and the evaluated SNPs of the *TLR9* gene, similar to those reported in the Brazil-French Guiana border [16]. However, the literature on this association varies according to the specific SNP analyzed. Regarding SNP rs5743836, a study conducted in Brazil reported an association between the

homozygous genotype for the minor allele (C/C) and high parasitemia [15]. Other studies reported an association between the homozygous genotype for the major allele (T/T) and low parasitemia [48,68]. These differences could be attributed to the genetic background of the population and the high genetic diversity of *Plasmodium*, which has multiple haplotypes and varied genetic patterns worldwide [14,16]. However, it has been shown that the homozygous genotype for the major allele (T/T) has higher promoter activity than the homozygous genotype for the minor allele (C/C) [54], potentially resulting in higher production of proinflammatory cytokines during malarial infection, favoring parasite control and elimination [48].

Several studies did not observe any association between SNP rs187084 and parasitemia [15,16,48]. However, one study in Brazil reported an association between the homozygous genotype for the major allele (C/C) and high parasitemia [68]. Peripheral parasitemia is only one indicator of disease state, but may not fully represent the total parasite burden, particularly in *P. vivax* infections where significant parasite biomass may be sequestered in organs such as the spleen and bone marrow [69]. It is plausible that this total biomass acts as a stronger driver influencing both the host's symptoms and the magnitude of the IgG response. Consequently, discrepancies in associations with peripheral parasitemia are perhaps expected. Moreover, numerous factors influence the clinical profile of the disease beyond parasite density, such as endemicity levels, multiplicity of infection [70], and the broader genetic composition of the human host [14,16]. These characteristics could act as confounding factors contributing to the variable findings observed between this and other studies on endemic malaria and should be considered when analyzing and interpreting the results [16]. No studies on the association between SNP rs352140 and parasitemia were found, suggesting that future research should investigate this.

We acknowledge limitations regarding our findings. Using symptom counts rather than severity grading (such as hemoglobin levels or splenomegaly quantification) impacts clinical interpretability. Financial and practical constraints prevented the collection of detailed hematological data (e.g., hemoglobin, platelets) and certain key symptoms (e.g., gastrointestinal distress, palpable splenomegaly), hindering a robust assessment of genotype-phenotype relationships. Viral co-infections present at the time of malaria infection are frequent in the study region [37], but were not assessed in the recruited individuals. Such co-infections could influence malaria symptomatology and immune responses, potentially confounding the relationships observed in our study. We also recognized that sample size and the focus on uncomplicated patients within a point-of-care setting limit generalizability. Future hospital- and community-based studies with larger, more diverse cohorts (including asymptomatic, mild, and severe cases) and comprehensive clinical/paraclinical assessments, including screening for relevant co-infections, are necessary to confirm our findings.

## Conclusions

Our findings suggest that genetic variation within the *TLR9* gene, specifically the rs5743836 (-1237T/C) heterozygous genotype, may modulate IgG antibody response against key *P. vivax* antigens: *Pv*CSP VK247 and V-like. Elucidating how host genetic factors shape pathogen-specific immunity is pertinent to the rational design of effective malaria vaccines. Future research should incorporate functional assays to determine the biological significance of the observed association between rs5743836 and IgG antibody response. Such studies should also measure total parasite burden and include asymptomatic and severe cases to assess the variant's impact across the *P. vivax* clinical spectrum. Longitudinal studies on populations with diverse malaria exposure and carrying the *TLR9*-1237T/C polymorphism may also reveal the risk of recurrent parasitemia and anemia, and the durability of specific IgG responses. Integrating standardized clinical evaluations within such longitudinal frameworks will be instrumental for establishing the contribution of *TLR9* gene polymorphisms to protective immunity against *P. vivax* malaria.

## Supporting information

**S1 Table. Peptide sequences corresponding to the three variants (VK247, VK210, and V-like) of the *Pv*CSP repetitive region.**
(DOCX)

**S2 Table. Description of polymorphisms, primer sequences, PCR amplification program, restriction enzymes, and fragments resulting from the genotyping of *TLR9* gene SNPs.**
(DOCX)

**S3 Table. Clinical-epidemiological characteristics of individuals infected with *P. vivax* by their IgG antibody response level against the VK247 *Pv*CSP variant.**
(DOCX)

**S4 Table. Clinical-epidemiological characteristics of individuals infected with *P. vivax* by their IgG antibody response level against the VK210 *Pv*CSP variant.**
(DOCX)

**S5 Table. Clinical-epidemiological characteristics of individuals infected with *P. vivax* by their IgG antibody response level against the V-like *Pv*CSP variant.**
(DOCX)

**S6 Table. Clinical-epidemiological characteristics of individuals infected with *P. vivax* by their IgG antibody response level against *Pv*MSP-1$_{19}$.**
(DOCX)

**S7 Table. Association of genotypic frequencies of *TLR9* gene SNPs with IgG antibody response level against the VK210 *Pv*CSP variant.**
(DOCX)

**S8 Table. Association of genotypic frequencies of *TLR9* gene SNPs with IgG antibody response level against *Pv*MSP-1$_{19}$.**
(DOCX)

**S9 Table. Association of haplotypic frequencies of *TLR9* gene SNPs with IgG antibody response level against the three *Pv*CSP variants and *Pv*MSP-1$_{19}$.**
(DOCX)

**S1 Dataset. Data that underlies this paper.**
(XLSX)

## Acknowledgments

We thank all volunteers who participated in this study and the technical staff involved in the samples collection.

## Author contributions

**Conceptualization:** Fhabián S. Carrión-Nessi, Mercedes Fernández-Mestre.

**Data curation:** Fhabián S. Carrión-Nessi, Eva Salazar-Alcalá, Kellys A. Curiel, Mercedes Fernández-Mestre.

**Formal analysis:** Fhabián S. Carrión-Nessi.

**Investigation:** Fhabián S. Carrión-Nessi, Eva Salazar-Alcalá, Kellys A. Curiel, Mercedes Fernández-Mestre.

**Methodology:** Fhabián S. Carrión-Nessi, David A. Forero-Peña, Mercedes Fernández-Mestre.

**Project administration:** Mercedes Fernández-Mestre.

**Resources:** David A. Forero-Peña, Kellys A. Curiel, Mercedes Fernández-Mestre.

**Supervision:** David A. Forero-Peña, Mary Lopez-Perez, Mercedes Fernández-Mestre.

**Validation:** Fhabián S. Carrión-Nessi, Eva Salazar-Alcalá, Mercedes Fernández-Mestre.

**Visualization:** Fhabián S. Carrión-Nessi, Mary Lopez-Perez.

**Writing – original draft:** Fhabián S. Carrión-Nessi, Eva Salazar-Alcalá, Kellys A. Curiel, Mercedes Fernández-Mestre.

**Writing – review & editing:** David A. Forero-Peña, Mary Lopez-Perez, Mercedes Fernández-Mestre.

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
