## [Decision Letter · Decision Letter 0]

TLR9 gene polymorphism -1237T/C (rs5743836) is associated with low IgG antibody response against PvCSP variants in a malaria-endemic area of Venezuela

Dear Dr. Carrión-Nessi,

Thank you for submitting your manuscript to PLOS Neglected Tropical Diseases. After careful consideration, we feel that it has merit but does not fully meet PLOS Neglected Tropical Diseases's publication criteria as it currently stands. Therefore, we invite you to submit a revised version of the manuscript that addresses the points raised during the review process.

Please submit your revised manuscript within 60 days May 08 2025 11:59PM. If you will need more time than this to complete your revisions, please reply to this message or contact the journal office at plosntds@plos.org. Please include the following items when submitting your revised manuscript:

We look forward to receiving your revised manuscript.

Kind regards,

Kevin Shyong-Wei Tan

Academic Editor

Guilherme Werneck

Section Editor

Shaden Kamhawi

co-Editor-in-Chief

Paul Brindley

co-Editor-in-Chief

**Additional Editor Comments:**

The manuscript has been assessed by three experts in the field, and while the results are interesting, the study lacks negative controls (asymptomatic cases) and functional validation. As such, the work cannot be accepted in its current stage and the major concerns of the reviewers need to be addressed, and the manuscript re-reviewed, before it can be considered for publication.

**Journal Requirements:**

At this stage, the following Authors/Authors require contributions: Fhabián S. Carrión-Nessi, Eva Salazar-Alcalá, David A. Forero-Peña, Kellys A. Curiel, Mary Lopez-Perez, and Mercedes Fernández-Mestre. Please ensure that the full contributions of each author are acknowledged in the "Add/Edit/Remove Authors" section of our submission form.

- TM on pages: 8, 9, and 10.

4) We have noticed that you have uploaded Supporting Information files, but you have not included a complete list of legends. Please add a full list of legends for your Data Set file after the references list.

**Comments to the Authors:**

**Please note that one of the reviews is uploaded as an attachment.**

**Reviewers' Comments:**

Reviewer's Responses to Questions

**Key Review Criteria Required for Acceptance?**

**Methods**

-Are the objectives of the study clearly articulated with a clear testable hypothesis stated?

-Is the study design appropriate to address the stated objectives?

-Is the population clearly described and appropriate for the hypothesis being tested?

-Is the sample size sufficient to ensure adequate power to address the hypothesis being tested?

-Were correct statistical analysis used to support conclusions?

-Are there concerns about ethical or regulatory requirements being met?

Reviewer #1: 1. Microscopy protocols require elaboration: Were thick smears examined by multiple microscopists? What quality assurance measures (e.g., cross-checking, WHO competency standards) were implemented? Parasitaemia distribution appears unusually constrained (median 5,880 parasites/μL, IQR 3,520–9,080). While high densities align with symptomatic presentations, broader ranges are typically observed (cf. Martins et al., 2015: median 4,200, IQR 1,400–18,000). Potential explanations (e.g., single-timepoint sampling, strict symptom-based enrolment) should be discussed.

2. The Clinical endpoints chosen have considerable limitations. The use of symptom count rather than severity grading (e.g., haemoglobin levels for anaemia, splenomegaly quantification) reduces clinical interpretability. Omission of haematological parameters (e.g., Hb, platelet count) and key symptoms (e.g., gastrointestinal distress, palpable splenomegaly) undermines the assessment of genotype-phenotype relationships. While the inverse correlation between antibody titres and symptom count is intriguing, its biological relevance remains unclear without severity metrics.

Reviewer #2: The objectives of the study are clearly articulated and the study design was appropriate to address these objectives.

A minor concern is that more details should be provided regarding the two-stage cluster analyses in which individuals were grouped into low, medium, and high responders according to their IgG levels (this was also done for associations with parasitemia). Please provide more details including which clustering methods were used and the refinement approach.

I was also curious about the study inclusion/exclusion criteria (Figure 1). Can the authors explain their rationale for excluding Caucasian and Amerindian individuals, and also state how Caucasian, Amerindian, and Venezuelan groups were defined?

In the methods, I might suggest that the authors explicitly state that all participants had uncomplicated malaria disease.

Reviewer #3: Suggestions are uploaded in document attachment.

**Results**

-Does the analysis presented match the analysis plan?

-Are the results clearly and completely presented?

-Are the figures (Tables, Images) of sufficient quality for clarity?

Reviewer #1: Results are presented well and the stats good

Reviewer #2: The results presented match the analysis plan and are clearly and completely presented.

Minor comment: Tables 2, 3, and 4 are a little hard to follow visually for the different models presented, would suggest editing to make it easier for the reader to see which rows align with the different models.

Reviewer #3: 1) Control groups, especially for the ELISA testing of IgG responses, need to be included to show specificity of assay.

2) ChemiDoc or Sanger sequencing results need to be shown for the 210 participants.

3) Neutralization capacity of antibodies need to be quantified.

Other suggestions are uploaded in document attachment.

**Conclusions**

-Are the conclusions supported by the data presented?

-Are the limitations of analysis clearly described?

-Do the authors discuss how these data can be helpful to advance our understanding of the topic under study?

-Is public health relevance addressed?

Reviewer #1: This study investigates how genetic polymorphisms in the TLR9 gene modulate immune responses to Plasmodium vivax malaria in Venezuela. The authors identified an association between the TLR9 -1237T/C variant (rs5743836) and reduced IgG antibody titers against two critical parasite antigens (PvCSP VK247 and V-like). Notably, no correlation was observed between this SNP and peripheral parasitemia levels—a finding consistent with the characteristically high parasite densities seen in symptomatic P. vivax infections. The focus on an understudied, high-risk population (Venezuelan miners) and the exploration of antibody-clinical symptom relationships are commendable strengths. However, the absence of functional validation data and exclusion of asymptomatic cases represent significant limitations.

The stated objective to “evaluate the frequency of SNPs in the TLR9 gene and their association with IgG antibody responses against PvCSP and PvMSP-119” is inherently confounded by exclusively enrolling microscopy-positive P. vivax cases. Given that subpatent infections dominate P. vivax epidemiology globally, the title and aims should explicitly clarify this limitation (e.g., “...in symptomatic, patent P. vivax infections”).

Reviewer #2: The conclusions are supported by the evidence presented. Limitations are described and include the sample size and study population being restricted to patients with uncomplicated disease. The authors do discuss how these results help to advance our understanding of how genetic variability may influence acquired immunity to malaria. Public health relevance is addressed.

Reviewer #3: Conclusions are weakly supported by data presented. Additional validations are required to strengthen and support the findings.

**Editorial and Data Presentation Modifications?**

Reviewer #1: (No Response)

Reviewer #2: (No Response)

Reviewer #3: Suggestions are uploaded in document attachment.

**Summary and General Comments**

Reviewer #1: I encourage the authors to consider longitudinal studies tracking how TLR9 -1237T/C influences recurrent infection risk, anemia development, and antibody durability would substantially strengthen these findings. Incorporating standardized clinical assessments and cellular immune readouts could further elucidate mechanisms.

Reviewer #2: In this study, the authors analyzed the association between three SNPs in the TLR9 gene and IgG responses to two Pv antigens among Venezuelan adults with acute uncomplicated Pv malaria. In this cohort, most (>90%) participants had positive IgG responses to PvCSP and PvMSP-119, and individuals with higher anti-PvCSP IgG levels had decreased symptomatology. The authors found that a heterozygous genotype for SNP rs5743836 was associated with lower IgG to two PvCSP peptide antigens.

The study helps to fill gaps in our knowledge regarding genetic polymorphisms and antibody responses to Pv, an area that is largely underreported compared to studies of Pf malaria. The manuscript is clearly written. The methods and statistical analysis are appropriate. The authors conclusions are well justified by the evidence presented.

Reviewer #3: Suggestions are uploaded in document attachment.

1) Additional malaria-naive negative controls for ELISA

2) Neutralization assay for antibodies against Malaria

PLOS authors have the option to publish the peer review history of their article (what does this mean? ). If published, this will include your full peer review and any attached files.

**Do you want your identity to be public for this peer review?** For information about this choice, including consent withdrawal, please see our Privacy Policy .

Reviewer #1: No

Reviewer #2: No

Reviewer #3: No

**Figure resubmission:**

**Reproducibility:**



---

## [Decision Letter · Decision Letter 1]

Dear Dr Fernández-Mestre,

We are pleased to inform you that your manuscript 'TLR9 gene polymorphism -1237T/C (rs5743836) is associated with low IgG antibody response against PvCSP variants in symptomatic P. vivax infections in Venezuela' has been provisionally accepted for publication in PLOS Neglected Tropical Diseases.

Best regards,

Kevin Shyong-Wei Tan

Academic Editor

Guilherme Werneck

Section Editor

Shaden Kamhawi

co-Editor-in-Chief

Paul Brindley

co-Editor-in-Chief

The authors have addressed the concerns of the reviewers satisfactorily.

Reviewer's Responses to Questions

**Key Review Criteria Required for Acceptance?**

**Methods**

-Are the objectives of the study clearly articulated with a clear testable hypothesis stated?

-Is the study design appropriate to address the stated objectives?

-Is the population clearly described and appropriate for the hypothesis being tested?

-Is the sample size sufficient to ensure adequate power to address the hypothesis being tested?

-Were correct statistical analysis used to support conclusions?

-Are there concerns about ethical or regulatory requirements being met?

Reviewer #1: (No Response)

Reviewer #2: The methods are appropriate and the authors have incorporated the reviewers' suggestions to better describe the methodology.

**Results**

-Does the analysis presented match the analysis plan?

-Are the results clearly and completely presented?

-Are the figures (Tables, Images) of sufficient quality for clarity?

Reviewer #1: (No Response)

Reviewer #2: In the revised manuscript, the authors' edits to the figures and table improve clarity.

**Conclusions**

-Are the conclusions supported by the data presented?

-Are the limitations of analysis clearly described?

-Do the authors discuss how these data can be helpful to advance our understanding of the topic under study?

-Is public health relevance addressed?

Reviewer #1: (No Response)

Reviewer #2: Revisions to the conclusions section improve the manuscript and more clearly describe limitations.

**Editorial and Data Presentation Modifications?**

Reviewer #1: (No Response)

Reviewer #2: Accept

**Summary and General Comments**

Reviewer #1: The authors have satisfactorily answered my questions

Reviewer #2: The revised manuscript incorporates the reviewers' suggested edits and is improved.

PLOS authors have the option to publish the peer review history of their article (what does this mean? ). If published, this will include your full peer review and any attached files.

**Do you want your identity to be public for this peer review?** For information about this choice, including consent withdrawal, please see our Privacy Policy .

Reviewer #1: **Yes: ** Bruce Russell

Reviewer #2: No

---

## [Editor Report · Acceptance letter]

Dear Dr Fernández-Mestre,

We are delighted to inform you that your manuscript, "TLR9 gene polymorphism -1237T/C (rs5743836) is associated with low IgG antibody response against PvCSP variants in symptomatic P. vivax infections in Venezuela," has been formally accepted for publication in PLOS Neglected Tropical Diseases.

Best regards,

Shaden Kamhawi

co-Editor-in-Chief

Paul Brindley

co-Editor-in-Chief
